



# Extension of the Langevin power curve analysis by separation per operational state

Christian Wiedemann[1,*], Henrik Bette[2,*], Matthias Wächter[1], Jan A. Freund[1], Thomas Guhr[2], and Joachim Peinke[1]

[1]Carl von Ossietzky Universität Oldenburg, School of Mathematics and Science, Institute of Physics, Oldenburg, Germany
[2]University of Duisburg-Essen, Fakultät für Physik, Duisburg, Germany
[*]These authors contributed equally to this work.

**Correspondence:** Christian Wiedemann (christian.wiedemann@uni-oldenburg.de)

**Abstract.** In the last few years, the dynamical characterization of the power output of a wind turbine by means of a Langevin equation has been well established. For this approach, temporally highly resolved measurements of wind speed and power output are used to obtain the drift and diffusion coefficients of the energy conversion process. These coefficients fully determine a Langevin stochastic differential equation with Gaussian white noise. We show that the dynamics of the power output of a

wind turbine have a hidden dependency on turbine's different operational states. Here, we use an approach based on clustering Pearson correlation matrices for different observables on a moving time window to identify different operational states. We have identified five operational states in total, for example the state of rated power. Those different operational states distinguish non-stationary behavior in the mutual dependencies and represent different turbine control settings. As a next step, we condition our Langevin analysis on these different states to reveal distinctly different behaviors of the power conversion process for

each operational state. Moreover, in our new representation hysteresis effects which have typically appeared in the Langevin dynamics of wind turbines seem to be resolved. We assign these typically observed hysteresis effects clearly to the change of the wind energy system between our estimated different operational states.

## 1   Introduction

Wind turbines have become a significant source of renewable energy due to their environmentally-friendly nature and po-

tential to generate electricity (Ackermann and Söder, 2002; Hannan et al., 2023). Analyzing their energy conversion process (Yaramasu et al., 2015) can be challenging due to the intricate nature influenced by various factors, including wind speed, turbulence, and mechanical wear. Nonetheless, a precise understanding of the dynamics of wind turbines is critical to simulate and consequently optimize their energy output and detect malfunctions (Wächter et al., 2011).

Recently, the Langevin equation approach has been used to study the dynamics of wind turbines (Tabar, 2019; Mücke et al.,

2015; Milan et al., 2010; Raischel et al., 2013; Lind et al., 2017). To capture the dynamics of the energy conversion process using this approach, highly resolved temporal measurements of wind speed and power output are employed to determine the drift and diffusion coefficients. These coefficients characterize both the deterministic and stochastic behaviors of the system.



However, this approach assumes a quasi-stationary system and does not consider the potential impact of different operational states.

We employ k means clustering, an established unsupervised machine method to the set of Pearson correlation matrices. This type of approach was first put forward in (Münnix et al., 2012) for financial correlation matrices, and is now also used in other fields such as traffic science (Wang et al., 2021, 2022, 2023). The method was further shaped and extended in (Heckens et al., 2020; Pharasi et al., 2020, 2021; Heckens and Guhr, 2022). Here, we employ recent progress on clustering structures for correlation matrices of wind turbines (Bette et al., 2021; Jungblut et al., 2022). By analyzing different observables over

a moving time window, they identified various operational states, which distinguish non-stationary behavior in the mutual dependencies and represent different turbine control settings. The dynamics of these operational states have been studied in (Bette et al., 2023), utilizing a Langevin ansatz to comprehend the deterministic behavior, enhancing the comprehension of the transition between these states.

This study employs the method of (Bette et al., 2021) to identify and distinguish various operational states of the wind

turbine, which are then used to condition the Langevin analysis. The analysis reveals unique behavior patterns in the power conversion process corresponding to each operational state and by that also successfully resolves hysteresis effects commonly observed in the power conversion process of wind turbines (Mücke et al., 2015; Lin et al., 2023).

## 2   Data set

The data utilized in this study is sourced from the Supervisory Control and Data Acquisition (SCADA) system of a Vestas

V90 turbine located in the Thanet offshore wind farm. These measurements were recorded at approximately 5-second intervals throughout the year 2017. To ensure consistent time stamps and a stable frequency, the data was aggregated by averaging over 10-second intervals. It is important to note that if no measurements were obtained within the original 5-second interval, the aggregated dataset may contain missing data during the corresponding 10-second interval. In this study we rescaled the ActivePower and the WindSpeed values.

The dataset under analysis comprises six variables, namely:

–   ActivePower: Generated active power

–   CurrentL1: Generated current (chosen from one of the three phases due to no deviations in the data)

–   RotorRPM: Rotations per minute of the rotor

–   GeneratorRPM: Rotations per minute of the high-speed shaft at the generator

–   BladePitchAngle: Blade pitch angle of the blades

–   WindSpeed: Wind speed



Our expectation for the V90 turbine is a shift in control strategy as the wind speed changes. This shift includes transitioning from a low wind speed regime with variable rotation speed and increasing power, to an intermediate regime with constant rotation and increasing power, and finally to a rated region with constant rotation and constant power. The selection of the

above six variables allows us to effectively track these operational state changes.

## 3    Correlation matrix states

In order to track the operational state of a turbine, we employ a method presented in Bette et. al. (Bette et al., 2021). Pearson correlation matrices are calculated for moving windows with non-overlapping time intervals called epochs to obtain a time series of correlation matrices. These are clustered to find structurally different operational states and thereby a time series $S(t)$,

which offers us the current operational state.

For this calculation we use all variables presented in sec. 2. Each of these variables is represented by a time series $X_k(t)$, where $k = 1, \ldots, 6$ represents the different variables and $t = 1, \ldots, T_{\text{end}}$ is the time variable. To capture non-stationarity we separate the whole time series into disjoint intervals $\lambda$ of length $T = 30\text{min}$. The interval length is chosen as a compromise between as short as possible to best resolve non-stationarity and as long as necessary to avoid noise in the correlation coefficients.

$\varepsilon$ represents the starting time of an interval $\lambda$.

Next, we normalize each time series in every interval to zero mean and standard deviation one by

$$\mu_k(t) = \langle X_k(t') | \varepsilon \le t' < \varepsilon + T \rangle \ , \ \ k = 1, \ldots, K, \ \ t \in \lambda \tag{1}$$

$$\sigma_k(t) = \sqrt{\left\langle \left(X_k(t') - \mu_k(t')\right)^2 | \varepsilon \le t' < \varepsilon + T \right\rangle} \ , \ \ k = 1, \ldots, K, \ \ t \in \lambda \tag{2}$$

$$G_k(t) = \frac{X_k(t) - \mu_k(t)}{\sigma_k(t)} \ , \ \ k = 1, \ldots, K \tag{3}$$

with $\mu_k(t)$ and $\sigma_k(t)$ being the mean value and the standard deviation of variable $k$ in the interval $\lambda$ with starting time $\varepsilon$. By arranging the variable time series in each epoch in a $K \times T$ data matrix

$$G(\lambda) = \begin{bmatrix} G_1(\varepsilon) & \ldots & G_1(\varepsilon + T - 1) \\ \vdots & & \vdots \\ G_k(\varepsilon) & \ddots & G_k(\varepsilon + T - 1) \\ \vdots & & \vdots \\ G_K(\varepsilon) & \ldots & G_K(\varepsilon + T - 1) \end{bmatrix} \cdot \tag{4}$$

we calculate the correlation matrix in the interval $\lambda$

$$C(\lambda) = \frac{1}{T} G(\lambda) G^{\dagger}(\lambda) \ . \tag{5}$$





Here, $G^\dagger(\lambda)$ denotes the transpose of $G(\lambda)$. Each matrix element $C_{ij}(\lambda)$ is the Pearson correlation coefficient between the variables $i$ and $j$ in the interval $\lambda$.

We apply hierarchical $k$-means clustering to find recurring states in our system. The algorithm is a divisive clustering that splits by applying standard $k$-means with $k = 2$. In each step, the cluster with the largest internal distance to its own center is split. Hence, we must define a distance $d(\lambda, \lambda')$ between the correlation matrices for intervals $\lambda$ and $\lambda'$:

$$d(\lambda, \lambda') = \sqrt{\sum_{i,j}(C_{ij}(\lambda) - C_{ij}(\lambda'))^2} = ||C(\lambda) - C(\lambda')|| \ . \tag{6}$$

The center of cluster $s$ is calculated as the element-wise mean $\bar{C}_{ij}(s) = \langle C_{ij}(\lambda)|S(\lambda) = s\rangle$. $S(t)$ is the function which results in the cluster $s$ for any time stamp $t$ assigned by the algorithm to the interval $\lambda$ that contains $t$. A more detailed description of the clustering procedure is found in Bette et. al. (Bette et al., 2021).

We visualize the different states of the power output for our dataset in figure **??**.

## 4    Estimation of the Kramers-Moyal coefficients


We start with the traditional approach to model the power conversion process

$$\dot{P}(t)|_{u(t)=u} = D_P^{(1)}(P(t), u) + \sqrt{D_P^{(2)}(P(t), u)} \cdot \Gamma(t) \tag{7}$$

of a wind turbine in terms of stationary Langevin equation (Risken and Risken, 1996; Wächter et al., 2011; Raischel et al., 2013; Tabar, 2019). Here, the power output $P(t)$ is modeled as a one-dimensional stationary stochastic process for a fixed wind

speed $u$. We assume a Gaussian distributed, delta correlated noise $\Gamma(t)$ with zero mean and a variance of 2.

Analytically, the n'th order conditional moments of the power output

$$M_P^{(n)}(P, u, \tau) = \langle (\Delta_\tau P(t))^n\rangle|_{P(t)=P, u(t)=u} \tag{8}$$

can be derived with expectation value of the increments to the power of $n$ $\Delta_\tau P(t)^n = (P(t+\tau) - P(t))^n$ over the time step $\tau$ at the specific state $(P, u)$ (Risken and Risken, 1996; Tabar, 2019).

With the n'th conditional moments, the n'th Kramers-Moyal coefficient

$$D_P^{(n)}(P, u) = \lim_{\tau \to 0} \frac{M_P^{(n)}(P, u, \tau)}{n! \cdot \tau} \tag{9}$$

can be calculated (Risken and Risken, 1996; Tabar, 2019).

We consider a two-dimensional dataset $(P, u)$ with $N$ datapoints with a uniform sampling interval $\tau_s = 1/f_s$. Furthermore, we define $\tau_m = m \cdot \tau_s$, where $m \in \mathbb{N}$. With this definition, we can calculate the increments of the power output over a time lag

$\tau_m$ with

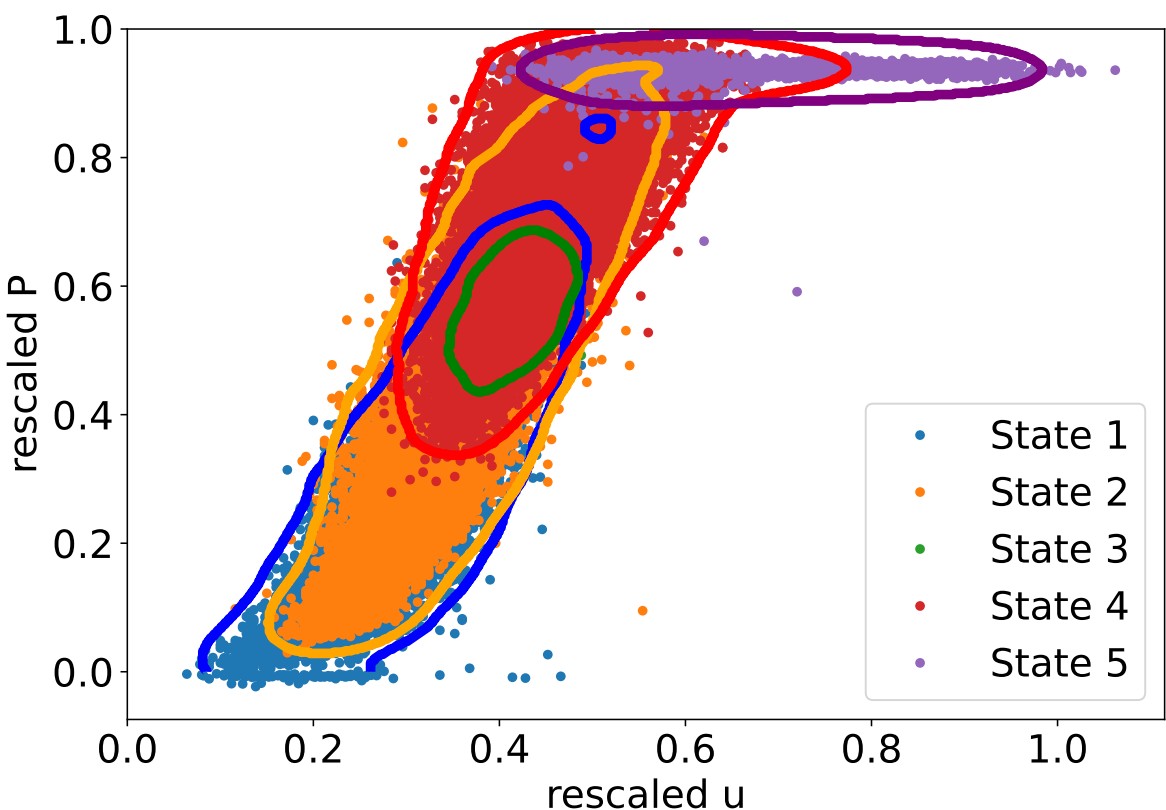

**Figure 1.** Visualizing of the data and the operational states: scatter plot with color-coded markers based on the operational state and contour lines based on the density (see appendix A) to show the borders of each operational state.

$$\Delta_{\tau_m} P_i = P_{i+m} - P_i. \tag{10}$$

To estimate the $n$-th conditional moment $M_P^{(n)}(P, u, \tau_m)$, we employ the Nadaraya-Watson estimator with a two-dimensional D-kernel $K_{a,b}(x, y) = k_a(x) \cdot k_b(y)$ (Nadaraya, 1964; Epanechnikov, 1969; Silverman, 1986). This D-kernel can be represented as the product of two one-dimensional kernels. For the first conditional moment, we can calculate the weighted average of the power output increments using

$$\hat{M}_P^{(n)}(P, u, \tau_m) = \sum_{i=1}^{N-m} (\Delta_{\tau_m} P_i)^n \cdot \frac{K_{a,b}\left(\frac{P_i - P}{h_P}, \frac{u_i - u}{h_u}\right)}{\sum_{j=1}^{N-m} K_{a,b}\left(\frac{P_j - P}{h_P}, \frac{u_j - u}{h_u}\right)}. \tag{11}$$





**Table 1.** Description of different kernel functions

| Name | Shape | Kernel function $k(x)$ with bandwidth $h$ |
|---|---|---|
| Rectangular | constant value within a fixed interval and drops abruptly to zero outside that interval | $\begin{cases} 1, & \text{if } \lvert x \rvert \leq h \\ 0, & \text{if } \lvert x \rvert > h \end{cases}$ |
| Gaussian | bell-shaped curve, characterized by a smooth and continuous decline in values away from the center | $\exp\!\left(-0.5 \cdot \left(\frac{x}{h}\right)^2\right)$ |
| Epanechnikov | flat and symmetric shape resembling a parabola, with its maximum value at the center and transitions to zero outside that interval | $\begin{cases} 1 - \left(\frac{x}{h}\right)^2, & \text{if } \lvert x \rvert \leq 1.0/h \\ 0, & \text{if } \lvert x \rvert > 1.0/h \end{cases}$ |

These weights in our specific case are determined by the states $P$ and $u$, as well as the kernel functions $k_a(x)$ and $k_b(x)$, along with the bandwidths for power output ($h_P$) and wind speed ($h_u$).

There are plenty of different kernel functions which are useful for different scenarios.

The Epanechnikov, Gaussian, and rectangular kernels are three commonly used kernel functions in non-parametric estimation and smoothing techniques (Epanechnikov, 1969). Each of these kernels has distinct properties that impact their use and the resulting estimation or smoothing outcomes. We summarized the three most commonly used kernels in table 1.

The choice between these kernels depends on the specific characteristics of the data and the desired properties of the estimation or smoothing procedure (Wied and Weißbach, 2012). The Epanechnikov kernel

$$k_{\mathrm{E}}(x) = \begin{cases} 1 - x^2 & , \lvert x \rvert \leq 1 \\ 0 & , \lvert x \rvert > 1) \end{cases} \tag{12}$$

is often favored when robustness and efficiency are important, and when a localized smoothing effect is desired. The Gaussian kernel is popular for its smoothness and computational efficiency. The rectangular kernel is suitable when simplicity and computational efficiency are prioritized, but it may not handle outliers or extreme values as the other kernels. In this study, we

use a Epanechnikov kernel function (12).

At least as important as the kernel function is the related bandwidth (Nadaraya, 1965; Jones et al., 1996; Scott, 2010; Silverman, 2018). For the analysis of large structures (macro-scale structures), large bandwidths should be used. However, with larger bandwidths, the small structures (micro-scale structures) are no longer visible. For a more comprehensive understanding of bandwidth selection for estimating Kramers-Moyal coefficients, we recommend consulting the study (Wiedemann et al.,

2024). To estimate the Kramers-Moyal coefficient of our specific dataset, we used the bandwidths according to the IEC 61400-





12-1 (IEC, 2005). The bandwidth $h_u$ for the wind speed is 1 m/s, and the bandwidth for the power $h_P$ is 100 kW. These bandwidths should be adjusted based on the given dataset (larger bandwidths for a smaller dataset, smaller bandwidths for a larger dataset). For the dataset we used, we found that these bandwidths, in conjunction with the Epanechnikov kernel, yield sensible results.

We assume that the $n$'th conditional moments $M^{(n)}(P, u, \tau)$ are linear for small time steps $\tau$. We can estimate the Kramers-Moyal coefficients

$$\hat{D}_P^{(n)}(P, u) = \frac{1}{M} \sum_{m=1}^{M} \frac{\hat{M}^{(n)}(P, u, \tau_m)}{n! \cdot \tau_m}. \tag{13}$$

by averaging the $n$'th conditional moments divided by the used (small) time step $\tau_m$ times $n$ factorial (Tabar, 2019). It can also make sense to give smaller $m$ values higher weight. We employed $M = 3$ for this particular dataset, yielding meaningful 135 outcomes.

We can determine the fixed points $P_0(u)$ of the system (Wächter et al., 2011). These fixed points correspond to values of $P$ at which the drift term becomes zero

$$\hat{D}_P^{(1)}(P_0, u) = 0, \tag{14}$$

indicating an equilibrium state.

Furthermore, in order to assess the stability of these fixed points, we examine the derivative of the drift at the fixed point. If the derivative is negative, it signifies that the fixed point is stable:

$$\frac{d}{dP} \hat{D}_P^{(1)}(P_0, u) < 0 \tag{15}$$

The derivative of the drift at the fixed point plays a crucial role in understanding the stability of the fixed point as well as providing valuable insights into the mean reversal time.

When studying the stability of a fixed point, we are interested in how the system responds to small perturbations of its equilibrium state. The derivative of the drift provides information about the local behavior of the system near the fixed point.

As said before, if the derivative of the drift evaluated at the fixed point is negative, it indicates that the fixed point is stable. In this case, any small disturbances from the equilibrium will eventually dampen out, and the system will return to its steady state. On the other hand, if the derivative is positive, the fixed point is unstable, and even the slightest perturbations will cause 150 the system to diverge from the equilibrium. Additionally, the derivative of the drift defines the mean reversal time of a system.

We make the assumption that the operational states $S(t)$ of the wind turbine can only take discrete values, specifically $S(t) \in [1, 2, 3, 4, 5]$ (which is related to the five identified clusters). Furthermore, we consider that both the drift and diffusion coefficients depend on the turbine's operational state. By incorporating this additional condition, we can reformulate the Langevin equation for the power conversion process using





$$\dot{P}(t)|_{u(t)=u, S(t)=S} = D_P^{(1)}(P(t), u, S) + \sqrt{D_P^{(2)}(P(t), u, S)} \cdot \Gamma(t). \tag{16}$$

The numerical approach can be derived in a similar manner as before. The only distinction is that we employ a 3-dimensional Kernel $K_{a,b,c}(x, y, z) = k_a(x) \cdot k_b(y) \cdot k_c(z)$. Due to the discrete values of the operational state, we can utilize a dedicated Boolean kernel function.

$$k_{\text{Bool}}(x) = \begin{cases} 1 & x = 0 \\ 0 & x \neq 0 \end{cases} \tag{17}$$

We use

$$\hat{M}_P^{(n)}(P, u, S, \tau_m) = \sum_{i=1}^{N-m} (\Delta_{\tau_m} P_i)^n \cdot \frac{K_{\text{a,b,bool}}\left(\frac{P_i - P}{h_P}, \frac{u_i - u}{h_u}, S_i - S\right)}{\sum_{j=1}^{N-m} K_{\text{a,b,bool}}\left(\frac{P_j - P}{h_P}, \frac{u_j - u}{h_u}, S_j - S\right)} \tag{18}$$

to estimate the n'th conditional moment at a specific state $(P, u, S)$. With these conditional moments we are able to obtain the Kramers-Moyal coefficients in a similar manner as shown above.

## 5 Stochastic analysis of the power conversion process

In this section, we elucidate the outcomes of our investigation into the wind turbine power conversion process using the Kramers-Moyal coefficients, considering both scenarios with and without separation per operational state. Our primary focus centers on analyzing the drift and diffusion values governing the power output of a wind turbine. To deepen our understanding, we extend the analysis to include the computation of fixed points, their associated stability and the diffusion values at these fixed points, revealing the nuanced dynamics intrinsic to diverse operational states.

The calculated drift values of the power output, as depicted in Figure 2, reveal a familiar pattern observed in prior studies without operational state separation (Wächter et al., 2011; Mücke et al., 2015; Milan et al., 2010). The top-left plot illustrates typical behavior in the power conversion process. Significant differences emerge when comparing drift maps for distinct operational states. Notably, a clear contrast is evident when analyzing State 2 against State 4, particularly at rescaled wind speeds ($u$) of approximately $0.4 - 0.5$. Similar variations are observed when comparing State 4 and State 5.

To deepen the analysis, we calculate stable fixed points and their derivatives. Figure 3 depicts stable fixed points per wind speed, highlighting disparities, particularly in regions characterized by rescaled $u$ values around $0.4 - 0.6$ across different operational states. Multiple stable fixed points are identified for a given wind speed, with States 1 and 2 displaying relative similarity. In contrast, significant differences are observed in other states, confirming the presence of hysteresis effects within the system dynamics (Mücke et al., 2015; Lin et al., 2023). The absence of multiple fixed points per wind speed without

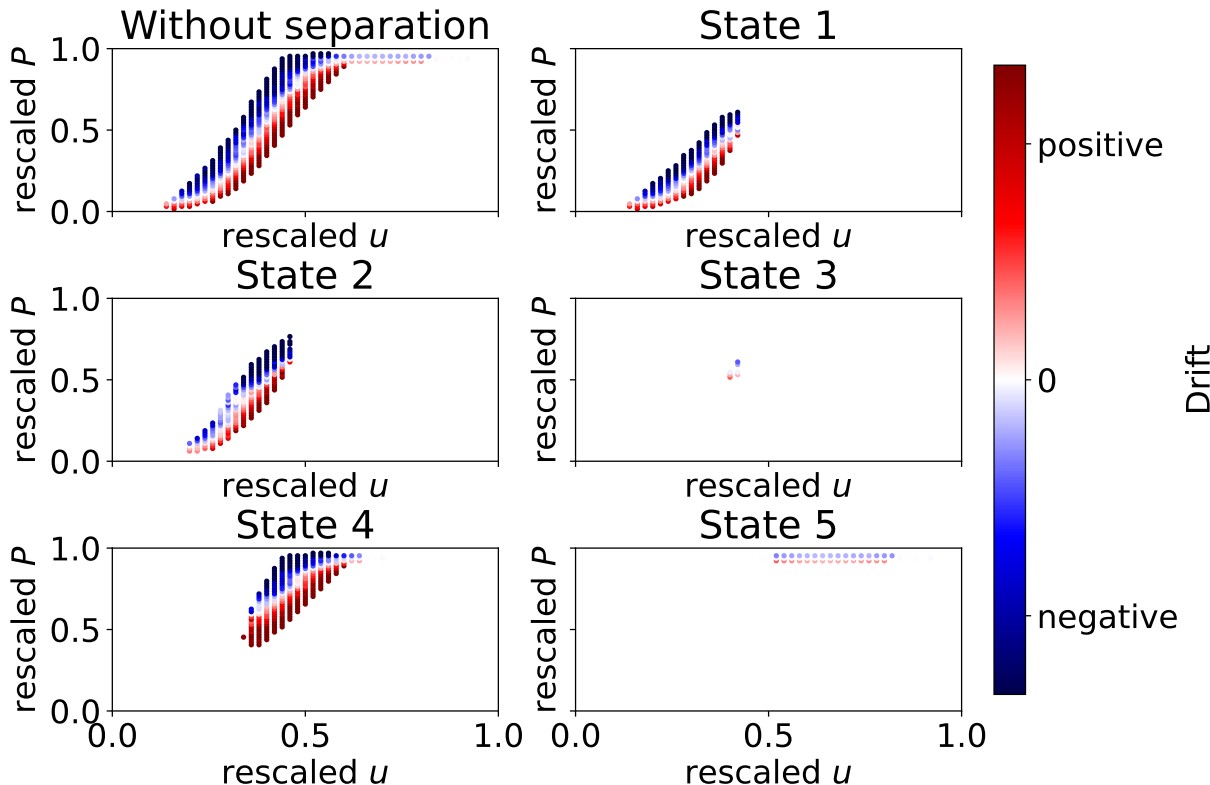

**Figure 2.** This figure illustrates the drift maps of the power conversion process, categorized by turbine states. The drift values are depicted using a color-coded scheme, with a corresponding colorbar provided on the right-hand side for reference.

operational state separation is attributed to the choice of a relatively high bandwidth during the estimation process for Kramers-Moyal coefficients. This, coupled with the use of a kernel function and the distribution of operational states, may have led to a more aggregated representation of the system dynamics.

We further explore the stability of these fixed points through the derivatives if the drift at the fixed points. Figure 4 illustrates the derivatives of the drift at stable fixed points per wind speed. Negative values signify stable fixed points, with larger absolute values indicating a shorter mean reversal time towards the fixed point. Comparing derivatives for different operational states reveals similarities for States 1, 2, 3, and 4 across rescaled $u$ values of approximately $0.0 - 0.6$. However, a significant change occurs for State 4 around $u \approx 0.6$, aligning it with State 5. Notably, at $u \approx 0.5 - 0.6$, clear differences emerge, with values for State 5 consistently lower than those for other states.

We extend our analysis from the deterministic parts of the behavior the calculation of the diffusion coefficients. The results are presented in Figure 5. Without operational state separation, diffusion values are generally smaller near fixed points than towards the edges. Small diffusion values are observed at rated wind speeds ($u > 0.5$) and lower power values ($P < 0.4$).



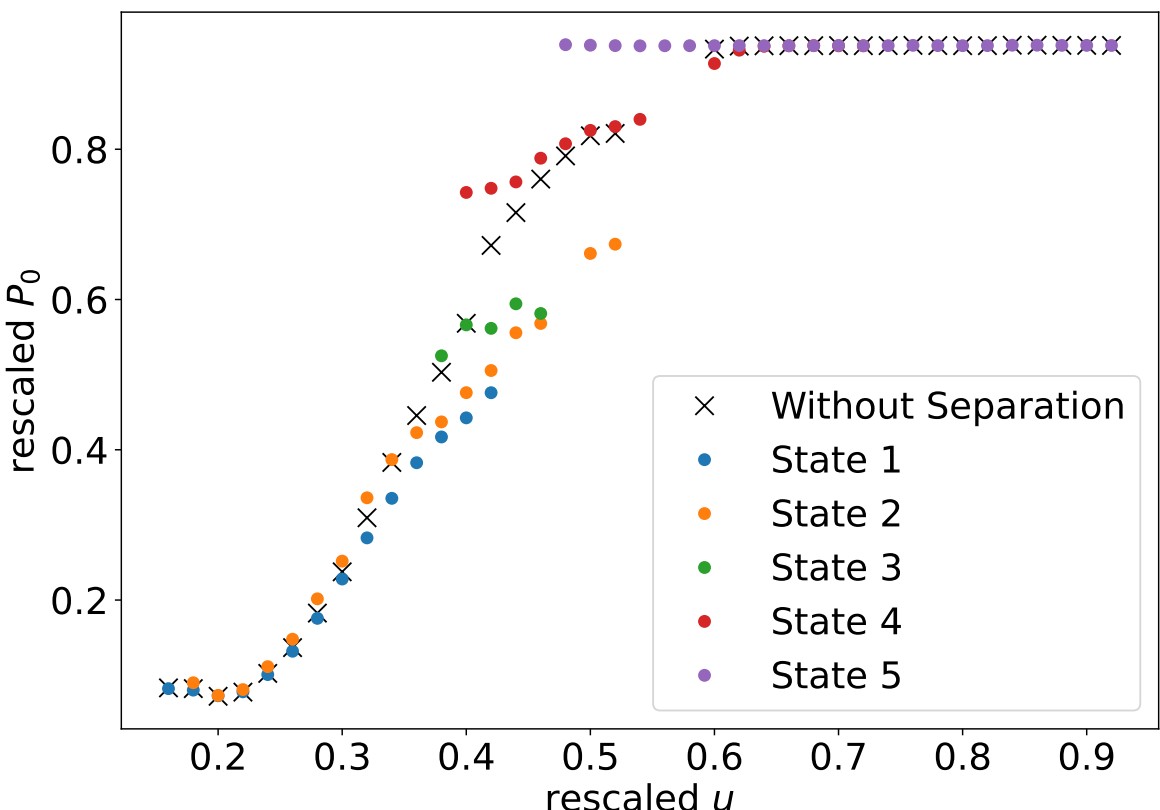

**Figure 3.** Rescaled stable fixed points $P_0$ of the power output in relation to the rescaled wind speed. The fixed points are categorized based on the operational states, with each state distinguished by a distinct color.

Differences between the diffusion values for different states are identified across various wind speeds, with States 1 and 2 exhibiting similarities, particularly around $u$ between $0.4$ and $0.6$.

Further analysis involves the calculation of diffusion values at the stable fixed points conditioned on wind speeds, visualized in Figure 6. Notably, here, we compare the diffusion values for the same wind speeds but different power values. States 1 and 2 exhibit similarity, while significant differences are observed across all other states, especially for $u$ between $0.4$ and $0.6$. The diffusion values for State 5 are smaller than those for other states at $u$ smaller than $0.6$.

The diffusion analysis further underscores distinctions in the behavior of State 5 compared to other states, particularly at wind speeds below $0.6$. The smaller diffusion values for State 5, coupled with the reduced derivatives at the fixed points, contribute to diminished fluctuations around these stable fixed points of the power time series of State 5 in comparison to other states at wind speeds below $0.6$. In contrast, State 2 exhibits higher diffusion values at the fixed points for wind speeds between

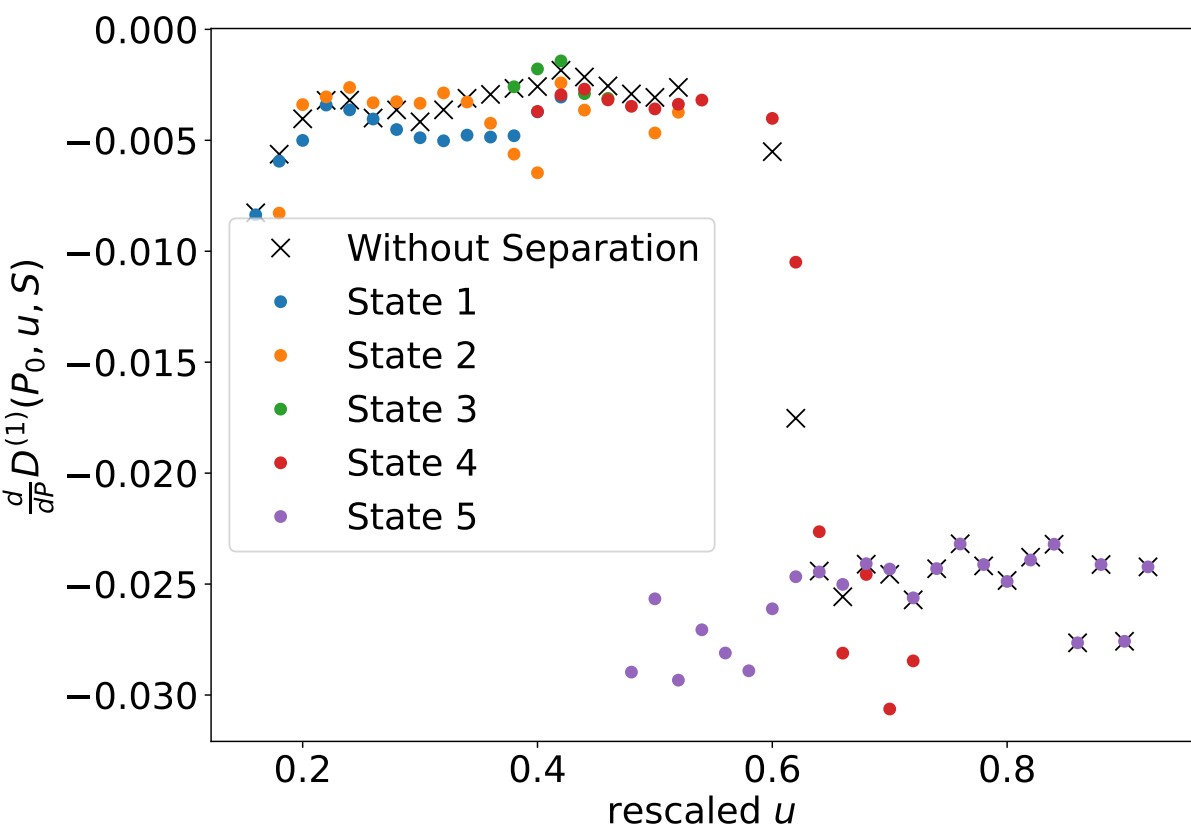

**Figure 4.** Derivative of the drift at the stable fixed points of the power output in relation to the rescaled wind speed. The derivatives are categorized based on the operational states, with each state distinguished by a distinct color.

0.4 and 0.55 than the other states. Having high diffusion values at the fixed points and similarities in derivative values with States 3 and 5 result in higher fluctuations for these wind speeds when compared to all other states.

## 6 Conclusions

In this study, we successfully applied a method to estimate the dynamics of the power conversion process while taking into account different operational states, identified using a correlation matrix algorithm (Bette et al., 2021) to take non-stationarity into account. Our analysis revealed distinct dynamics associated with each operational state in the power conversion process, emphasizing the significant influence of these states on the overall behavior of the system.



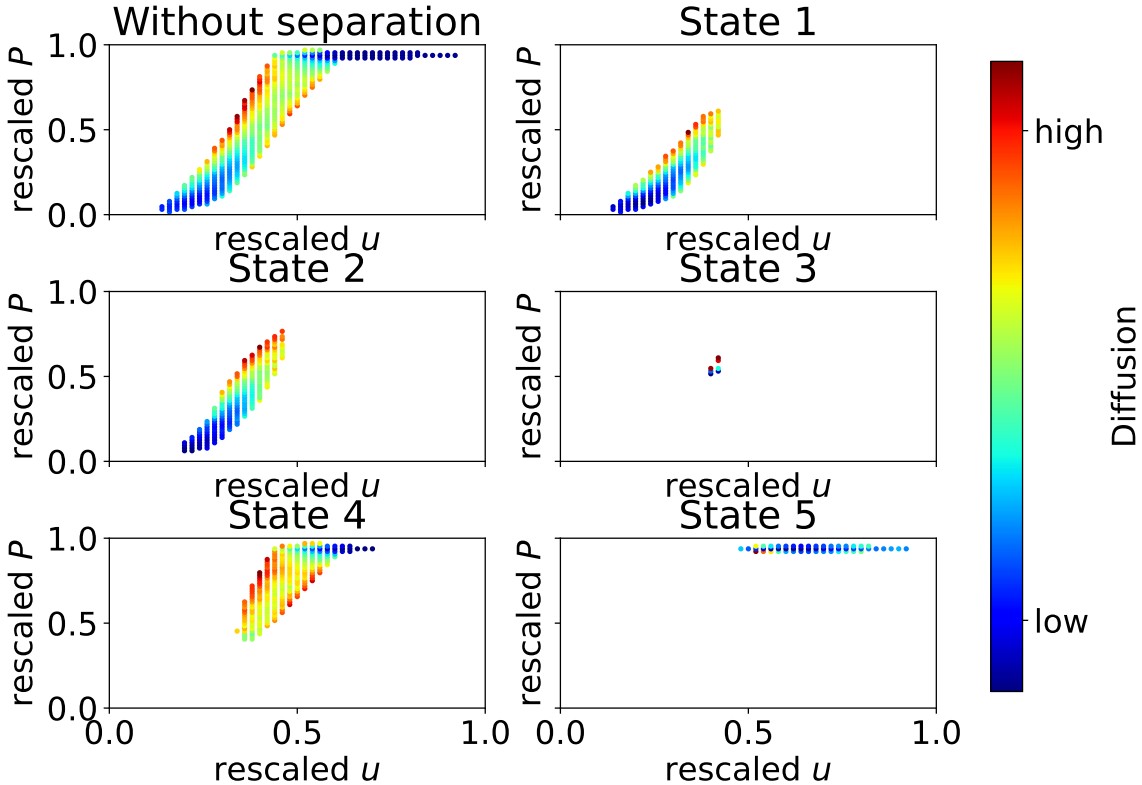

**Figure 5.** Diffusion maps of the power conversion process, categorized by turbine states. The diffusion values are depicted using a color-coded scheme, with a corresponding colorbar provided on the right-hand side for reference.

We successfully resolved hysteresis effects within the power conversion process. When separating per operational state

distinct fixed points per wind speed are visible. Without accounting for states, these are averaged out into one fixed point per wind speed. The presented analysis also allows to identify differences in the dynamic behavior of states. State 5, representing rated power production, displayed a much more stable behavior with less fluctuations than other states. This remained true even for wind speed values where State 5 overlaps with other states.

The results clearly show that it is possible to enhance existing methods by considering the described operational states.

The analysis concept does not need to change much, but rather only takes the automatically detected operational state as a distinction parameter for multiple subanalyses with the original method.

**Appendix A:  Operational state contour lines**

To represent the distribution of operational states visually, we utilize kernel density estimation given by:





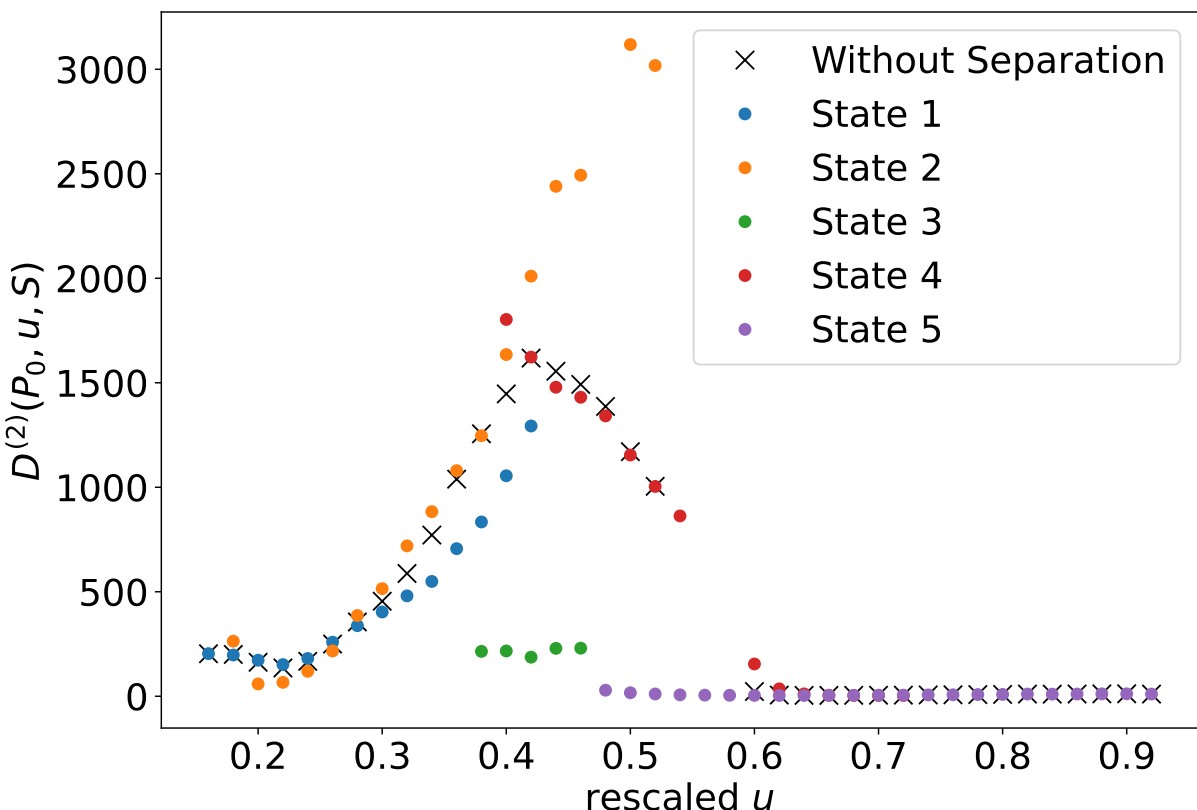

**Figure 6.** Diffusion values at the fixed points of the power output in relation to the Rescaled wind speed. The diffusion values are categorized based on the operational states, with each state distinguished by a distinct color.

$$\hat{f}(u_0, P_0, S_0) = \sum_{i=1}^{N} K_{\text{gauss,gauss,bool}} \left( \frac{P_i - P_0}{h_P}, \frac{u_i - u_0}{u_h}, S_i - S_0 \right). \tag{A1}$$

Contour lines are then generated using the formula:

$$\tilde{f}(u_0, P_0, S_0) = \begin{cases} 1 & , \rho(u_0, P_0, S_0) \geq \rho_0 \\ 0 & , \text{else} \end{cases} \tag{A2}$$

Here, $\rho_0$ is a predefined threshold (set at 20).





*Data availability.* The data that support the findings of this study are available from Vattenfall AB. Restrictions apply to the availability of these data, which were used under license for this study. Data are available from the authors with the permission of Vattenfall AB.

*Author contributions.* C.W. and H.B. conceived and developed the approach, J.A.F., M.W., T.G., and J.P. contributed to its formalization. All authors contributed to the writing of the manuscript and approved it.

*Competing interests.* At least one of the (co-)authors is a member of the editorial board of Wind Energy Science.

*Acknowledgements.* We extend our sincere gratitude to *Vattenfall AB* for generously providing the data essential for this study. We also wish to acknowledge the valuable insights gained from discussions with David Bastine and Timo Lichtenstein. The research presented here was

conducted as part of the *Wind farm virtual Site Assistant for O&M decision support – advanced methods for big data analysis* (WiSAbigdata) project, funded by the Federal Ministry of Economics Affairs and Energy, Germany (BMWi). We express our appreciation to BMWi for their financial support, which greatly facilitated this work.





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
