# Peer review of "Extension of the Langevin power curve analysis by separation per operational state"

_Wind Energy Science, 2024_

## Author Comment (AC1)

**1 Answers to the comments of reviewer 1**

The paper shows that the power output dynamics of a wind turbine have a hidden dependency on the turbine's different operational states. By identifying these states using a correlation matrix clustering method, the authors were able to condition the Langevin analysis on the different states. This revealed distinct power conversion behaviors for each state and resolved previously observed hysteresis effects, which were attributed to changes between the operational states. The results emphasize the importance of accounting for the different states to accurately capture the complex dynamics of the wind turbine power generation process.

The message of the manuscript is very interesting and sound. Although the paper is well-written and extremely timely, there are some improvements to consider before final acceptance in WES.

**1.1 Add a small subsection about hysteresis effects and how the new analysis is resolving it.**

**Answer**  We agree that adding a subsection on hysteresis effects would enhance the quality of the paper. Therefore, we have included a dedicated subsection to discuss these effects and how the new analysis addresses them.

**Section 2, new in the revised manuscript**  Hysteresis effects are a significant phenomenon observed in the operational dynamics of wind turbines, particularly evident in high frequency SCADA data during the power conversion process. These effects arise primarily from two interrelated factors: the switching of operational states and the stochastic nature of wind.

In order to ensure the safe operation of wind turbines, control strategies are implemented which actively regulate the power output and trigger different operating states depending on the most recent history of the same wind speed. To avoid instability in the view of rapid fluctuations robust control strategies induce delayed switching between a finite number of operating states.

The switching of operational states, which occurs in response to varying wind conditions, leads to a nonlinear response in turbine performance. For instance, when wind speed fluctuate, turbines may transition between modes such as cut-in, rated, and cut-out states. This switching can result in distinct performance paths during wind acceleration and deceleration. Consequently, the system exhibits hysteresis, where the output power does not return along the same trajectory when wind conditions reverse, leading to discrepancies in expected versus actual power generation.

In our analysis, we specifically condition the dynamics on the operational state rather than examining the transitions between them. This approach is crucial, as it reveals that the dynamics of the power conversion process are intrinsically linked to the operational state in which the turbine is functioning. Without conditioning on operational states, the analysis of the dynamics of the power conversion process yields a mixed average of dynamics across different

states, effectively diluting the unique characteristics associated with each operational condition. By separating the data according to operational states, we can derive the distinct dynamics pertinent to each state. This granularity enables us to identify multiple stable fixed points at various wind speeds, corresponding to different operational states. Importantly, this methodology clarifies the hysteresis effects induced by the switching of operational states. By isolating each state, we can observe how these stable fixed points shift in response to changes in wind conditions, thereby resolving the complexities of hysteresis that may otherwise obscure our understanding.

**1.2 Change the naming of the indexing variable k in Eqs. (1)-(3) to avoid any confusion with the k-means clustering method. Unify the notations used for k-mean, k mean.**

**Answer**   Thank you for bringing this to our attention. The notation for the k-means has been unified by changing the parameter from $k$ to $l$.

**Section 4, new in the revised manuscript**

$$\mu_l(t) = \langle X_l(t') | \varepsilon \leq t' < \varepsilon + T \rangle \quad , \quad l = 1, \ldots, L, \quad t \in \lambda \tag{1}$$

$$\sigma_l(t) = \sqrt{\left\langle \left(X_l(t') - \mu_l(t')\right)^2 | \varepsilon \leq t' < \varepsilon + T \right\rangle} \quad , \quad l = 1, \ldots, L, \quad t \in \lambda \tag{2}$$

$$G_l(t) = \frac{X_l(t) - \mu_l(t)}{\sigma_l(t)} \quad , \quad l = 1, \ldots, K \tag{3}$$

**1.3 Define the diffusion coefficients after Eq. (7) and Eq. (16) to provide a clear explanation of these important parameters.**

**Answer**   We added a description for these important parameters.

**Section 5, new in the revised manuscript**   In the context of stochastic differential equations, the ensemble dynamics of a stochastic process can be described using the Kramers-Moyal coefficients. The first coefficient represents the drift, which indicates the average tendency of the system to move in a certain direction. This drift reflects trends that would be considered as the backbone in a deterministic model approach. The second coefficient corresponds to the diffusion, capturing the variance of the random fluctuations or noise in the system. This diffusion coefficient quantifies how spread out the possible states of the process can become over time due to these random influences.

**1.4 Describe how the optimal bandwidth h of the kernel is estimated, as this can significantly impact the results.**

**Answer** In the revised manuscript, we have clarified our explanation regarding the choice of bandwidth in section 5. The challenge lies in calculating the optimal bandwidth in both 2- and 3-dimensional spaces. Instead of using a specific method to compute the optimal bandwidth, we experimented with a range of different bandwidths and selected those that yielded fully reliable results.

**1.5 Write a short note about the possibility that the diffusion term may change the location of the stable fixed points obtained from the drift term, leading to noise-induced transitions.**

**Answer** Since the noise is symmetric, we do not consider noise-induced drift in our model. Noise-induced transitions can occur due to the stochastic nature of the system, such as wind fluctuations, which can contribute to hysteresis effects. We added a note to the revised manuscript in section 2

**Section 2, new in the revised manuscript** The random fluctuations of the wind can trigger noise-induced transitions, causing the system to shift from one stable state to another, even when the drift term indicates stability in the original state.

**1.6 Discuss the impact of potential jumps in the power output that may be present in the different operational states S=1,...,5.**

**Answer** This is an interesting and important topic. However, we are unsure whether the observed jumps in power output are present within a single operational state or only occur during transitions between different states. It is important to note that only the controller is capable of reacting quickly enough to induce visible jumps in the data. Yes, it is important to better understand the dynamics of power output fluctuations across operational states. This, however, requires quite a bit of further research which is beyond the scope of the present paper.

**1.7 Unify the citation style used throughout the References section.**

**Answer** Thank you for your comment. We have carefully reviewed the References section and ensured consistency in the citation style throughout the manuscript.

**2  Answers to the comments of reviewer 2**

The authors present a framework to derive operational states of one wind turbine, based in (linear) correlation metrics of six observables describing the turbine's behavior. Combining these matrices with a k-means algorithm to cluster them, the author identify 5 operational states with distinct dynamical features.

I found this idea interesting to explore frameworks to analyse wind turbine behavior. However, I would suggest to expand the main text to discuss it. In particular, I have the following remarks:

**2.1  At least a more torough description of the parameter when applying the framework should be provided. Namely, how is T determined? Is there an optimal value. balancing trade-off between accuracy and statistical uncertainty?**

**Answer**  Thank you for your comment. We agree that a more thorough description of the parameter $T$ is necessary.

**Section 4, new in the revised manuscript**  The 30-minute time span represents a compromise. The choice of $T$ depends on the specific system being analyzed, as it must balance two competing factors: accuracy and statistical uncertainty. A larger $T$ improves statistical reliability by averaging over more data points, reducing noise. However, longer time windows can obscure short-term dynamics and fail to capture rapid changes in the system. Conversely, smaller $T$ values provide higher temporal resolution but may introduce greater statistical uncertainty due to fewer data points. The 30-minute time span represents a compromise. In our approach, we select $T$ based on a trade-off that best captures the system's dynamics while maintaining a reasonable level of statistical accuracy. Given that external factors, such as wind, can change on timescales ranging from minutes to hours, shorter epochs are required to capture the non-stationarity. Such trade-offs are common when working with time series of correlation matrices.

**2.2  How would such an approach scale with the number of turbines, i.e. applied to a wind farm? Would it enable to also derive large scale operational states of wind farms?**

**Answer**  We thank the reviewer for the interesting hint.

**Section 4, new in the revised manuscript**  An open question is the possible existence of operational states of complete wind farms, which could be investigated following a similar methodology. The results could potentially support the operation of wind farms, namely wind farm control.

**2.3 The use of linear correlations (only) seems odd in the context of a (highly) non-linear system. I am aware of the previous works by Guhr and co-workers with correlation matrices between stocks, but I wonder why e.g. mutual information or even granger causality matrices were not consider. At least some content at the level of discussion would be in order.**

**Answer**   We intentionally chose a simple first-order approach to see if it could already provide meaningful insights – and it has indeed done so. In a next step, more refined methods such as mutual information or Granger causality could be explored to gain additional insights.

**Section 4, new in the revised manuscript**   While we focused on linear correlations in this study, which offer a simple and effective first-order approach, we acknowledge that the system's non-linearity may suggest the potential for more advanced methods. Techniques such as mutual information or Granger causality could provide deeper understanding of the relationships between variables.

**2.4 The authors should in the end make a careful proof-reading, to detect several typos, missed references ("??") and improve the layout of some tables and figures.**

**Anwer**   Thank you for your valuable feedback. We will carefully proofread the manuscript to identify and correct any typographical errors, missing references ("??"), and improved the layout of the tables and figures as suggested.